# The Effects of a 4-Week Combined Aerobic and Resistance Training and Volleyball Training on Fitness Variables and Body Composition on STEAM Students

Martin Pacholek [1,2,]*[ ], Erika Zemková [3,4][ ], Keith Arnolds [5] and Peter Šagát [1]

1   Health and Physical Education Department, Prince Sultan University, Riyadh 12435, Saudi Arabia; sagat@psu.edu.sa
2   Physical Education Department, College of General Studies, King Fahd University of Petroleum & Minerals, Dhahran 31261, Saudi Arabia
3   Department of Biological and Medical Sciences, Faculty of Physical Education and Sports, Comenius University in Bratislava, 81469 Bratislava, Slovakia; erika.zemkova@uniba.sk
4   Faculty of Electrical Engineering and Information Technology, Sports Technology Institute, Slovak University of Technology in Bratislava, 81219 Bratislava, Slovakia
5   Prep-Year Program Department, College of General Studies, King Fahd University of Petroleum & Minerals, Dhahran 31261, Saudi Arabia; kvarnolds@kfupm.edu.sa
*   Correspondence: mpacholek@post.cz; Tel.: +96-65-0200-2099

**Abstract:** The study evaluates the effects of a 4-week program of combined resistance and aerobic training and volleyball training on physical fitness in young sedentary adults. Twenty-eight males (age $20.5 \pm 1.5$ years; body mass $87.2 \pm 28.5$ kg; height $173 \pm 8.1$ cm; BMI $28.9 \pm 8.4$ kg/m$^2$) were divided into two groups. While experimental group 1 (COM) underwent a fitness program consisting of a combination of strength and aerobic exercises, experimental group 2 (VOL) performed intermittent exercises in volleyball (four times per week for 50 min). The results showed that both training groups significantly improved in all fitness tests except the beep test, while only the COM group achieved a significant change in the number of repetitions ($p = 0.041$). Between-group analyzes revealed a greater change achieved by the COM group in standing long jump lengths than VOL (12.3% vs. 4.3%, $p = 0.011$). There were no other between-group significant differences in 20m Multistage Fitness Test (8.1% and 4.4%,), sit-ups (20.9% and 21.0%), flexibility (24.5% and 23.3%) and shuttle run $5 \times 10$ m (11.95 and 9.52%) or in anthropometric parameters (BMI, Fat %, Muscle Mass %, Visceral Fat %). These findings indicate that combined resistance and aerobic training are more effective in improving the explosive power of lower limbs and aerobic endurance than playing an intermittent-type sport such as volleyball. This program can be effectively applied to university students with a predominantly sedentary lifestyle.

**Keywords:** fitness program; physical education; motivation; concurrent training

## 1. Introduction and Literature Review

The limitation of time and number of classes demands teachers to use the most effective methods and exercises that have beneficial effects on students' health and future motivation to be physically active. Teachers should always look for new methods, approaches on how to motivate and activate their students for better performance and results during the physical education (PE) classes.

Strength and endurance training produce significantly different or even opposite outcomes. Aerobic endurance training reduces the activity of glycolytic enzymes and increases the number of intracellular energy storages, the activity of oxidative enzymes, the number of capillaries in muscles, and the density of mitochondria [1]. Strength training, on the other hand, has nearly opposite results to these factors, although both training types increase the number of intracellular energy storages. Moreover, endurance training mostly

maintains or even decreases the size of muscle fibers whereas strength training increases them [2]. Overall, endurance training increases aerobic processes, and strength training increases muscle strength, anaerobic processes, and power production. Applying a short but intensive strength training program (up to 6 weeks) can lead to significant changes in some parameters in strength performance, especially in overall strength and neural adaptation. On the other hand, it is not possible to achieve any significant changes in muscle hypertrophy [3].

Combining these two training methods, strength and endurance training (commonly known as concurrent training), appears to be quite efficient since most of the studies on athletes regarding combined training have led to positive results in both strength and endurance performances [4,5]. McCarthy et al. noticed in their study that the combined strength and endurance training group gained as much strength as the group that did only strength training; the first group also improved their maximal oxygen uptake as much as the group that did only endurance training [6]. From *A Meta-Analysis Examining the Interference of Aerobic and Resistance Exercises*, it is known that the total power is the major variable affected by concurrent training, and it does not compromise muscle hypertrophy and maximal strength development [7]. However, explosive strength gains may be attenuated [8]. Moreover, studies found that combination training positively affects the maximal strength of lower limbs in untrained and moderately trained individuals [9,10]. Other studies found protentional health benefits on cardiovascular, neuromuscular, hormonal, immunological, virological, and body composition parameters. These changes in the combined training program had more or less the same benefit compared with just one of these individually [11]. The American College of Sports Medicine recommends combining aerobic and resistance training for its advantages in contrast to strength or endurance training alone [12]. Concurrent training has not been fully investigated, and there are still many questions. The effect of this training is a multidimensional phenomenon, influenced by many physiological and nonphysiological factors such as type of exercise, training background, muscle groups involved, and inter-individual variations [13].

Volleyball training is commonly used to improve skill-related aspects of the game while also enhancing health-related fitness variables. Studies that investigate the effect of volleyball training on students found higher aerobic capacity [14,15], better explosive strength, muscular endurance, flexibility [16], functional strength, speed, coordination skills, elasticity, and balance [17] with low risk of injury; this makes volleyball an effective tool in physical education classes. Nevertheless, according to Trajkovič, a limited number of studies have investigated the effects of small-sided games in volleyball on physical fitness in children and adolescents [18].

The novelty of the present study is that previous studies [6,8] compared the effects of concurrent strength (S) and endurance (E) training between each other or the order effect (S + E, E + S) [19–22]. This study compares physical variables between concurrent and volleyball training in a homogenous group of university students during physical education classes. For this reason, the main aim of this study was to evaluate and compare the effects of an intensive workout program combined with an aerobic training program (in each PE class) with classical volleyball classes on selected physical fitness variables and anthropometric abilities. We wanted to incorporate a combined training into PE classes because of the benefits of this kind of stimulus. Both activate different kinds of muscle fibers and affect the cardiovascular system differently. This effect may postpone fatigue, prolong physical activity and increase the effectiveness of PE classes. These results may be valuable for effective planning, building, and execution of PE classes with an overall goal to increase the general fitness of students.

It was hypothesized that the training program (combination aerobic and strength exercises) will enhance performance in strength and endurance significantly more than the volleyball program and that we will not record any significant changes in subjects' anthropometric abilities after both tested groups underwent four weeks of intensive programs.

This study was based on a combination of aerobic and strength training in the same PE class. The purpose of this study was to observe the effects of an intensive 4-week resistance (four workout programs) and aerobic training program on selected physical fitness variables of STEAM University students and compare these results with a group that underwent a volleyball training program.

## 2. Materials and Methods

### 2.1. Participants

Twenty-eight physically inactive male university students were randomly chosen to act as subjects for the study. Only male students were selected because the university did not accommodate female students at that time. The medical history of participants was taken into consideration in order to select healthy and non-injured participants exclusively. None of the students had any experience with resistance training nor aerobic training in the past. The first experimental group ($n = 14$; age = $20.5 \pm 1.5$ years; weight = $98.7 \pm 27.4$ kg; height = $177.5 \pm 6.3$ cm) underwent a fitness program in which every class was divided into aerobic and resistance training (COM). The second experimental group ($n = 14$; age = $20.2 \pm 1.2$ years; weight = $75.6 \pm 25.6$ kg; height = $168.4 \pm 7.3$ cm) underwent volleyball classes (VOL). All students completed a 4-week intensive training program. The procedures followed the ethical standards on human experimentation stated in compliance with the 1964 Helsinki Declaration and its later amendments. The project was approved by the ethics committee of the Deanship of Scientific Research at King Fahd University of Petroleum and Minerals in Dhahran (SB181037).

### 2.2. Study Design

The study was designed to investigate and compare the effects of four-week training programs (combination of strength-aerobic training and volleyball classes) on STEAM untrained university students from Saudi Arabia. The study was a double group–time parallel experimental research design with a gradual independent variable characterized by fitness programs and the dependent variable characterized by levels of physical fitness and anthropometric abilities. The first experimental factor was a combination program that utilized resistance training stimuli such as isometric, working out on machines, HIT (high-intensity training), dynamic with equipment, and aerobic training stimuli. The second experimental factor was the volleyball program. The training period for the study was four days per week for four weeks. In one physical education class (50′), two different stimuli (20′-aerobic runs together with coordination and 20′-strength training) were applied. For both training programs, fitness tests and anthropometric measurements were observed and implemented with one instructor for better validity of results.

### 2.3. Training Programs

The training period for the study was four days per week for four weeks (16 × training—each PE class had a duration of 50 min) without any food restriction. A recommendation was provided about foods and drinks that students should not have during the training period, specifically mentioning, fast food, sweets, sugary drinks, and fruit juices with added sugar.

- Combination of Aerobic and resistance training program

Each aerobic and resistance training consisted of four different parts (5′-introduction, 20′-aerobic runs with different variations of exercises, 20′-strength training, and 5′-cooldown). Four different stimuli for the aerobic and resistance part of the training were applied, which changed based on the day of the week (Table 1). A working week in Saudi Arabia is from Sunday to Thursday. During the introduction to each session, it was briefly explained to students what the main part of each PE class would consist of. The first five minutes of the aerobic part of the class was used for a proper warmup with different agility exercises around cones (Figure 1). For the next 15 min, jogging with coordination exercises were used on Sunday, jogging with jumping and athletics drills

on Monday, running in different competitive games like hunting and relay on Wednesday, and running in intermittent loads around the indoor field (based on students' fitness and subjective feelings) on Thursday. Afterward was the resistance part of the PE class, beginning with an explanation of strength exercises which students had time to try in a limited number of repetitions (1–3×) as a warmup for the first set of the workout part. Then, participants underwent isometric resistance training on Sunday, which contained eight exercises (plank, wall sit, contralateral limb raise, static abdominal crunch, side plank, curtsy lunge, superman, and abdominal leg raise). On Monday, students worked out in the gym, and the program contained these exercises: bench press, sit-ups on a decline bench, hyperextension, leg press, pectoral fly, abdominal flexion, row machine, and squats on the bench. Wednesday was for Tabata training which included four exercises: jumping jacks, burpees, high knees, mountain climbers and on the last day, students underwent resistance training with equipment (expander snap, rolling on foam roller, throwing a medicine ball, farmer's walk with dumbbells, expander raise, side to side with a medicine ball, push-ups around an obstacle, jump on the bench or step on it). The students worked out based on their subjective feelings with maximal effort based on repetition, time, weight of resistance, or based on the instructor who was implementing individualized training programs based on students' fitness abilities. At the end of the class, compensation and stretching exercises were done with encouragement from the instructors. During the strength training, the circle organization form was applied because of limited time during classes.

**Table 1.** Training protocol for aerobic and resistance training program.

| Class Days | Sunday | Monday | Wednesday | Thursday |
|---|---|---|---|---|
| **Aerobic training (20′)** | Jogging + Coordination | Jogging + Athletics drills | Jogging + Games | Jogging |
| Rest interval | Individual | Individual | Individual | Individual |
| Load | Intermittent | Intermittent | Intermittent | Intermittent |
| **Resistance training (20′)** | Isometric training | Gym training (machine) | Tabata training | Gym training (equipment) |
| Rest interval | 15–30″ | 1:1 | 10″ | 20″ |
| Load interval | 30–45″ | - | 20″ | 20″ |
| Number of repetitions | - | 12–20× | 8× | Individual |
| Number of sets | 3× | 2× | 3 | 3 |
| Rest interval between the sets | 90″ | 90″ | 2 | 90″ |
| Number of exercises | 8 | 8 | 4 | 8 |

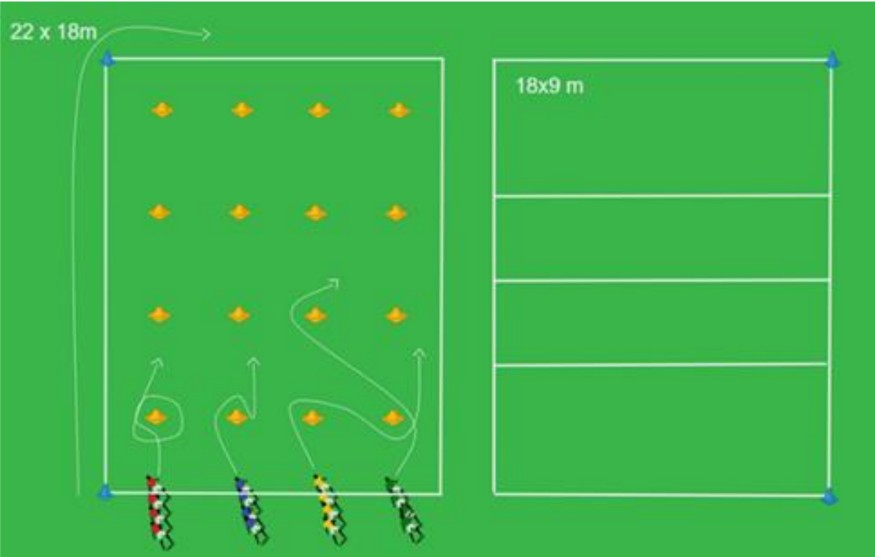

**Figure 1.** The aerobic program.

- Volleyball Training Program

The volleyball class consisted of four parts. In the introduction part (5′), the main aim of the PE class was explained with an explanation of the correct technique for individual volleyball skills. Students focused on different skills (overhead pass, forearm pass, service, and smash) each week. After the introduction, a warm-up (5′) with jogging and dynamic stretching followed. The main part was divided between the training of skills (20′) in isolated conditions and small-sided games (2 vs. 2, 3 vs. 3) on a court size (3–4.5 m in width, 6–9 m in length) which focused on these skills. The main part of every PE class ended with a volleyball game (15′). At the end of each class, compensation exercises (5′) were performed to cool down.

### 2.4. Fitness Tests and Anthropometric Measurements

Pre- and post-fitness and anthropometric tests were taken one week before and after the training period. Five fitness tests measuring a different physical ability were chosen. The measurement was scheduled for two PE classes, and it always started with a general warm-up (10′) and finished with compensation exercises (5′). On the first day, different anthropometric abilities such as fat %, muscle mass, body weight, visceral fat, and body mass index (BMI) were tested. The biometric scale TANITA DC 430MA (Tanita Corporation, Tokyo, Japan) was used for this measurement. Then, for examination of hamstrings and back muscles flexibility, one attempt of the sit and reach test was performed; this was followed by a standing long jump (SLJ) to determine the explosive power of lower limbs. The last fitness test in the first PE class was the shuttle run test ($10 \times 5$ m), which measures the speed and agility of students. The time was measured with the FITRO Gates photocell device (FITRONIC, Bratislava, Slovakia). The next class measured the strength of abdominals and hip flexor muscles using the sit-up test (30″) and cardiovascular fitness level together with maximum oxygen uptake (VO2 max) by the beep test. The methodology of selected fitness tests was taken from the Eurofit fitness testing battery [23].

### 2.5. Statistical Analysis

Measurement-testing and content analysis were used for obtaining data, and a questionnaire, mathematical, statistical, and logical methods were used. Data are presented as means, percentages, and standard deviations (SD). The Shapiro–Wilk test for normality was performed on all variables. Data showing a normal distribution paired sample t-test was used to find significant differences in pre- to post-tests from selected fitness tests in the aerobic and resistance training group and volleyball group. For non-normal distribution data, the Wilcoxon signed-rank test was used. Two-Sample *t*-test and Mann-Whitney test were used for comparing the results of both groups with each other. The level of significance was set at $p \leq 0.05$, and statistically significant differences were marked with a symbol (*). In order to interpret the practical significance of the research results, the effect size (ES) was reported. Cohen defined effect sizes as small, d = 0.2; medium, d = 0.5; and large, d = 0.8 while Pearson defined correlation r = 0.1 to 0.3 as small, r = 0.3 to 0.5 as medium, and r = 0.5 to 1.0 as large [24]. Statistical and data analyses were performed using the statistical program IBM SPSS Statistics 21 (IBM Corporation, Armonk, NY, USA).

## 3. Results

As shown in Table 2, both groups (VOL and COM) improved significantly in all selected fitness tests except the beep test, which recorded improvements only in the combination group ($p = 0.05$). A comparative group analysis showed significantly greater improvements in COM than VOL in the length of standing long jump (12.3% vs. 4.3%, $p = 0.011$). However, there were no other significant between-group differences.

**Table 2.** Analysis of data for the selected fitness variables from Pre- to Post-test in Combination and Volleyball group.

| Fitness Variables | Pre-Test | | Post-Test | | Percentages (%) | p Value | Effect Size |
|---|---|---|---|---|---|---|---|
| | Mean | S.D | Mean | S.D | | | |
| **Sit ups test (repetitions)** | | | | | | | |
| Volleyball group | 21.40 | 3.74 | 25.90 | 4.50 | 21.03 | 0.001 | r = 0.74 |
| Combination group | 19.10 | 4.81 | 23.10 | 4.39 | 20.94 | 0.000 | d = 0.87 |
| **Flexibility (cm)** | | | | | | | |
| Volleyball group | 23.60 | 6.71 | 29.10 | 6.00 | 23.31 | 0.000 | d = 0.86 |
| Combination group | 18.80 | 8.14 | 23.40 | 9.95 | 24.47 | 0.005 | d = 0.51 |
| **Shuttle run 5 × 10 m (s)** | | | | | | | |
| Volleyball group | 14.70 | 1.20 | 13.30 | 1.13 | −9.52 | 0.002 | r = 0.59 |
| Combination group | 15.90 | 1.66 | 14.00 | 1.02 | −11.95 | 0.000 | d = 1.38 |
| **Standing long jump (cm)** | | | | | | | |
| Volleyball group | 176.0 | 25.99 | 183.5 | 28.30 | 4.26 | 0.001 | d = 0.28 |
| Combination group | 153.1 | 20.69 | 171.9 | 23.92 | 12.28 | 0.000 | d = 0.84 |
| **Beep test (stages)** | | | | | | | |
| Volleyball group | 58.90 | 21.97 | 61.50 | 23.94 | 4.41 | 0.146 | |
| Combination group | 56.71 | 17.57 | 61.29 | 17.33 | 8.08 | 0.041 | d = 0.26 |

From the anthropometric measurement (Table 3), it was found that only the combination group significantly decreased the percentage of visceral fat ($p = 0.05$). The other anthropometric variables did not change significantly. The comparative analysis does not show any significant differences between groups.

**Table 3.** Analysis of data for the selected anthropometric variables from Pre- to Post-test in Combination and Volleyball group.

| Fitness Variables | Pre-Test | | Post-Test | | Percentages (%) | p Value | Effect Size |
|---|---|---|---|---|---|---|---|
| | Mean | S.D | Mean | S.D | | | |
| **BMI** | | | | | | | |
| Volleyball group | 26.5 | 7.71 | 26.2 | 7.45 | −1.13 | 0.529 | |
| Combination group | 31.3 | 8.61 | 31.3 | 8.36 | 0 | 0.892 | |
| **FAT %** | | | | | | | |
| Volleyball group | 24.5 | 9.12 | 24.1 | 9.01 | −1.63 | 0.820 | |
| Combination group | 27.3 | 9.90 | 27.9 | 9.39 | 2.20 | 0.876 | |
| **Muscle Mass %** | | | | | | | |
| Volleyball group | 65.7 | 11.12 | 65.4 | 10.99 | −0.46 | 0.802 | |
| Combination group | 66.9 | 10.32 | 66.8 | 10.09 | −0.15 | 0.912 | |
| **Visceral Fat %** | | | | | | | |
| Volleyball group | 9.34 | 7.03 | 9.28 | 6.98 | −0.64 | 0.724 | |
| Combination group | 9.79 | 7.27 | 9.50 | 7.09 | −2.96 | 0.040 | d = 0.04 |

## 4. Discussion

The purpose of this study was to examine and discover the effects of four-week training programs during physical education classes on anthropometric and fitness variables. Combination training consisted of two blocks of training (aerobic, resistance) in each PE class and the volleyball program consisted of isolated skill exercises, small-sided games, and volleyball games. The main findings of our research were that both groups (COM, VOL) significantly improved in selected fitness tests, except for the endurance (beep) test, where only the COM group significantly improved (8.08%). These results confirm the hypothesis that the COM group improved their performance not only in strength but also in endurance compared with the VOL group. Furthermore, the COM group achieved significantly greater improvements in the explosive power of lower limbs (standing long

jump) than the VOL group (12.28% vs. 4.26%). No other significant changes were noticeable after the comparative analysis. The anthropometric abilities stayed significantly unchanged in both groups after the selected training program. The only exception was visceral fat in the COM group, where the subjects significantly decreased their percentage of fat.

Combining these two training methods, strength and endurance training, was also effective in different studies [5,25,26]; most studies regarding combined training have led to positive results in both strength and endurance performance. McCarthy et al. noticed in their study that the combined strength and endurance training group gained as much strength as the group that performed only strength training. They also improved their maximal oxygen uptake as much as the group that did only endurance training [5]. These findings correlate with the results of this study, where the COM group improved significantly in all measured fitness variables after the selected period of four weeks.

Generally, positive changes in fitness variables (endurance, flexibility, agility, strength, and explosive power) on students after volleyball intervention were noticed in many studies [16–18,27] but also in many group sports during physical education classes [28,29]. Significant improvement of maximum oxygen uptake does not appear in the VOL group, which could have happened because of insufficient running activity during the volleyball training program. However, this does not correspond with the results of Mohammed, who found a significant change in time (10.1%) running 1.6 km rafter the volleyball program, which was conducted twice a week for eight weeks [16]. An interesting finding came from the comparative analysis for explosive strength of lower limbs when the COM group achieved significantly better improvements than the VOL group. Similar results were achieved by Trajkovič when the volleyball group improved significantly after 8 months of intervention (twice a week) in vertical jump, but this change was not significant compared with the control group [18]; Sozen also did not find any improvement in broad jump by high school students (untrained vs. volleyball group) in his research [28]. These studies confirm the fact that combination training is more effective for improving explosive strength of lower body muscular power.

Studies related to anthropometric abilities found that combined training caused a decrease in fat levels around the abdominal area and combined training was more effective than food intake limitations [30]. Using combined aerobic and strength exercise was the most effective training program in obesity training [31]. A different study showed that training should be medium or long term to achieve positive effects. Short-term training in less than 12 weeks has no effect on decreasing body mass [32], but some research found significant results even in a shorter period. Park's study showed that physical parameter results of combined training conducted for an 8-week long training protocol positively affected body weight, body mass index, body fat percentage, and maxVO2 values in women [33]. However, Schumann did not find significant changes in body fat and blood lipids even after a 24 weeks' intervention (2–3× per week) on physically active men, but found increasing total lean mass [21], which also corresponds with the study of Lee [20].

An eight weeks' study on junior volleyball players (3× per week) had similar results where no significant differences between pre-training and post-training for body mass or skinfold thickness were found [34]. On the other hand, Mohammed found significant changes in body mass and BMI for university students [16]. From all these results, it is unclear how much effect the combination program has on anthropometric parameters, and it appears that many factors (gender, age, physical activity, duration) can influence changes in anthropometric variables. Results from this study can confirm that 4-week training programs (COM, VOL) do not show any significant changes in BMI, fat percentage, or muscle mass percentage on untrained male students. Only the percentage of visceral fat significantly improved in the COM group during the selected program. The short period and lack of food restrictions during the training programs may have influenced the final anthropometric results of this study.

The major limitation of this short four-week study is that it involved a small group of students without a control group. A further limitation relates to the COM group being an

average of 23.1 kg heavier than the VOL group, which may have influenced the fitness and anthropometric results. Moreover, the calorie intake of students was not controlled during the four weeks. In future research, it would be interesting to compare these results with other team sports, particularly with participants who are also required to have strength and endurance abilities like basketball and football. Furthermore, the use of select fitness tests specific to the differentiation between these two programs can be explored. Changing the length, intensity, and training stimuli may lead to better efficiency of these programs and variability of results.

## 5. Conclusions

This research provides valuable information on how the combination program can be effectively applied to physical education classes and how it can influence the level of physical fitness compared with the volleyball program. Based on the results, it may be concluded that the selected programs (COM and VOL) were effective for changing physical fitness variables, but COM was more effective in improving explosive strength of the lower body, and showed a significant change in the number of repetitions during the beep test compared with the VOL program. Both programs were ineffective in changing anthropometric abilities, but the COM group significantly improved in the % of visceral fat. The COM program is effective and applicable during PE classes with a limited amount of time and space, and the students can receive necessary theoretical information and practical experience from different methods of endurance and strength training.

**Author Contributions:** Conceptualization, M.P., E.Z. and K.A.; methodology, M.P.; formal analysis, M.P.; investigation, M.P.; data curation, M.P., E.Z. and K.A.; writing—original draft preparation, M.P.; writing—review & editing, M.P., E.Z. and K.A.; visualization, M.P.; supervision, E.Z.; project administration, K.A.; funding acquisition, M.P., E.Z. and P.Š. All authors have read and agreed to the published version of the manuscript.

**Funding:** This work was supported by the Scientific Grant Agency of the Deanship of Scientific Research, King Fahd University of Petroleum and Minerals in Dhahran (SB181037) and the authors would like to recognize the efforts made by Prince Sultan University in funding the research either with fees, incentives, or seed grants.

**Institutional Review Board Statement:** The procedures followed the ethical standards on human experimentation stated in compliance with the 1964 Helsinki Declaration and its later amendments. The project was approved by the ethics committee of the Deanship of Scientific Research at King Fahd University of Petroleum and Minerals in Dhahran (SB181037; 14 February 2020).

**Informed Consent Statement:** Informed consent was obtained from all subjects involved in the study and written informed consent has been obtained from the patients to publish this paper.

**Data Availability Statement:** The data presented in this study are available on request from the corresponding author. The data are not publicly available due to privacy reason here.

**Conflicts of Interest:** The authors declare no conflict of interest.

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
