# Peer review of "The Effects of a 4-Week Combined Aerobic and Resistance Training and Volleyball Training on Fitness Variables and Body Composition on STEAM Students"

_applsci, doi:10.3390/app11188397_

Round 1

Reviewer 1 Report

the authors responded partial to my comments, now the article looks much better, as I said even if there are no significant differences between the two groups it is very important to show that what the experimental group used is better compared to the control group , that would have been the idea.

My big question for the authors is why there are no differences between the two groups given that the content of the workouts was different, here is a big dilemma for me. When you talk about something new, you should get results, right?

References are not as required by the journal, please see the instructions for authors.

Author Response

Dear Reviewer,

  • We express our gratitude for the valuable feedback provided in your review reports. Your recommendations greatly improved our study and also enhanced our knowledge of best practices in the field. We trust that the affected changes meet your satisfaction.

Comments and Suggestions for Authors

the authors responded partial to my comments, now the article looks much better, as I said even if there are no significant differences between the two groups it is very important to show that what the experimental group used is better compared to the control group , that would have been the idea.

My big question for the authors is why there are no differences between the two groups given that the content of the workouts was different, here is a big dilemma for me. When you talk about something new, you should get results, right?

  • The fact is that both of the programs were effective but almost in all tests there were no significant difference between them. It could have happened because of the selected tests (could have been more specific) or maybe the program wasn’t long enough to record changes between these two programs. Maybe different training stimuli or changing the intensity of the programs could lead to better efficiency of these programs and variability of results.

References are not as required by the journal, please see the instructions for authors.

  • We have changed some references and put them on the end of the sentences.

Reviewer 2 Report

I congratulate the authors for the work done. After reviewing the manuscript, I have observed that some changes are necessary.

1.- The next sentence should go to the end of the introduction section:

“This study was based on the combination of aerobic and strength training in the same PE class, so the purpose of this study was to observe the effects of an intensive four-week resistance (four workout programs) and aerobic training program on selected physical fitness variables of STEAM University students and compare these results with a group which underwent a volleyball training program”

2.- The following sentence must be associated with a study and must be cited:

“Aerobic endurance training reduces the activity of glycolytic enzymes, but increases the number of intracellular energy storages, the activity of oxidative enzymes, and the number of capillaries in muscles and the density of mitochondria”

3.- The numbers that indicate the citations must go at the end of the sentence:

  • [5]: line 61.
  • [6]: line 64.
  • [8,9]: line 68.
  • [23]: line 211.

4.- The sit up test does not assess the hip flexors, this test assesses the hip extensors (line 195).

5.- I consider that they should include studies in which the physical condition is evaluated using the Alpha batteryand not the Eurofit fitness battery.

- Field-based fitness assessment in young people: the ALPHA health-related fitness test battery for children and adolescents.

6.- Table 2 should indicate the measure that has been used for the Fitness variables.

Author Response

Dear Reviewer,

  • We express our gratitude for the valuable feedback provided in your review reports. Your recommendations greatly improved our study and also enhanced our knowledge of best practices in the field. We trust that the affected changes meet your satisfaction.

I congratulate the authors for the work done. After reviewing the manuscript, I have observed that some changes are necessary.

1.- The next sentence should go to the end of the introduction section:

“This study was based on the combination of aerobic and strength training in the same PE class, so the purpose of this study was to observe the effects of an intensive four-week resistance (four workout programs) and aerobic training program on selected physical fitness variables of STEAM University students and compare these results with a group which underwent a volleyball training program”

  • We have put this sentence at the end of the introduction section.

2.- The following sentence must be associated with a study and must be cited:

“Aerobic endurance training reduces the activity of glycolytic enzymes, but increases the number of intracellular energy storages, the activity of oxidative enzymes, and the number of capillaries in muscles and the density of mitochondria”

  • We have cited this paragraph in the text

3.- The numbers that indicate the citations must go at the end of the sentence:

  • [5]: line 61.
  • [6]: line 64.
  • [8,9]: line 68.
  • [23]: line 211.
  • All these references were moved to the end of the sentence

4.- The sit up test does not assess the hip flexors, this test assesses the hip extensors (line 195).

  • We have found in literature that:
  • sit-up tests measure the strength and endurance of the abdominals and hip-flexor muscles
  • Burden AM, Redmond CG. Abdominal and hip flexor muscle activity during 2 minutes of sit-ups and curl-ups. The Journal of Strength & Conditioning Research. 2013 Aug 1;27(8):2119-28.

5.- I consider that they should include studies in which the physical condition is evaluated using the Alpha batteryand not the Eurofit fitness battery.

- Field-based fitness assessment in young people: the ALPHA health-related fitness test battery for children and adolescents.

  • We have used literature where authors used the same or similar fitness tests but also the ALPHA fitness test battery contains the 20 m shuttle run test, standing broad jump and BMI which was used in our test battery.
  • We have added a study which used the Alpha health related fitness test battery on university students to the discussion.

6.- Table 2 should indicate the measure that has been used for the Fitness variables.

  • We have added all measure units to each test into Table 2.

Reviewer 3 Report

There is a problem with the sample; significant differences in the average weight and height between the members of the two compared groups at the beginning of the study. According to the BMI results in Table 3, authors had an overweight group (VOL) and an obese group (COM) who also underwent different programmes; the VOL group aerobic, while the COM group aerobic and anaerobic programme. Maybe a 4-week tested period, although undoubtedly useful for the participants, was not long enough to detect a reduction in body fat and an increase in muscle mass? In the volleyball group, the percentage of body fat was already acceptable before the programme started (and the big change was not expected especially because no nutrition intervention was made), while for the combination group 16 exercise sessions in 4 weeks were not enough for changes in the neuro hormonal system (again, with no food or drink restrictions).

Abstract

1st sentence, I propose rephrasing into: "The study evaluates the effects of 4-week programs of combined resistance and aerobic training, and volleyball training on the physical fitness of young adults with sedentary lifestyle"

line 15, quantitative measures of BMI, height, and body mass should have the same number of decimal places for mean and sd

line 19, put a comma in front of word "while"

line 21, reorder into: "standing long jump compared to VOL (12.3% vs 4.3%, p=0.011)"

Introduction

line 62, it does not fit to have two times word "that", could you change it into "... that the total power ..."?

line 80, a comma is missing after "[15]"

line 87, there is an extra space after the fullstop

line 100, please add "in subjects` anthropometric abilities"

line 101, please change into "after both tested groups went through ..."

Materials and Methods

Participants

lines 109-110, 112; weight and height should have the same number of decimal places for mean and sd

line 113; change into "completed a 4-week intensive programe" or "completed 4-weeks of"

It would be worth mentioning somewhere that a work week in Saudi Arabia is from Sunday to Thursday.

Results

line 217; I think that there is an extra space after the fullstop

Table 3; two p values have two decimal numbers, while others have three

Conclusions

Although in general I agree with the authors that a combination training is better than the volleyball training, I am not sure that results of this study support such a conclusion; there were two different groups based on their nutritional status (overweight and obese) who underwent different types of exercise programmes and yet in both groups statistically significant changes were detected.

Author Response

Dear Reviewer,

We express our gratitude for the valuable feedback provided in your review reports. Your recommendations greatly improved our study and also enhanced our knowledge of best practices in the field. We trust that the affected changes meet your satisfaction.

We have changed all these mistakes and added information about working days in KSA.

Round 2

Reviewer 1 Report

  • The authors say: ''It could have happened because of the selected tests (could have been more specific) or maybe the program wasn’t long enough to record changes between these two programs. Maybe different training stimuli or changing the intensity of the programs could lead to better efficiency of these programs and variability of results''........there may be a limit to the research if the authors did not show significant differences........... as justified in my comment

References: please put the abbreviation for all journals

Author Response

Dear Reviewer,

We have changed shortcuts in the references section and added information to the limitation of the study.

Thank you again for your help and feedback.

This manuscript is a resubmission of an earlier submission. The following is a list of the peer review reports and author responses from that submission.

Round 1

Reviewer 1 Report

I apologize, I am unclear what the overall goal is for this study. Sometimes it seems the goal is to improve explosive power and perform at a high rate of intensity.  Other times it seems the goal is to find the activity that is most enjoyed (and will be sustained) by the members. 

The two groups are referred to as "experimental group 1 and 2" , "control versus experimental", "COM and VOL" and a lot of statements about "the experimental group" - I'm not clear on which group is the experimental group... volleyball? Best to be consistent and use the same group name throughout.

First paragraph introduction, volleyball not even mentioned in the purpose of the study. 

The whole section on feedback confused me.  I'm not really sure how it fits into the paper - is it that doing these suggestions would allow teachers to get better feedback?

Main aim of the study, like 86, I believe this is the first time student perception is mentioned?

Participants: your two study groups are 25 kg different, could this change the results per group?

Study design, how can this study be double blinded? The participants know whether they are playing volleyball or doing resistance training. 

I am not clear on the part where the subjects set up their personal goals, this needs to be explained in further detail.  Did they have a list of goals to choose from? So this is what they were evaluating the program based on, in the end?

You mention providing nutrition recommendations, I think that information needs to be included. 

For the resistance training, was there a set intensity? Line 130 "work out based on their subjective feelings in the selected range..." were they told to be at a specific intensity?

Table 2, there are two r= in the effect size.  Should these be d= ?

Table 4, for the questionnaire about the selected programs, did the participants have to rank the 5 programs, only using 1, 2, 3, 4, 5 each one time per question?  Like only one of the programs could get ranked 5?
It seems like it based on the fact that all of the data is whole numbers, and 1-5 are each only used once per question?

Did volleyball participants get to fill out a questionnaire ranking their program?  Did they write down their goals at the beginning and assess the effectiveness of the program also?

Reviewer 2 Report

The study seeks to present the assessment and effects of the combined 4-week endurance and aerobic volleyball training program on physical fitness variables in sedentary adults. To improve the article I would have some suggestions for authors:

  1. Introduction - The authors must present several current studies close to this topic, also must be mentioned in this chapter and what is the novelty of the study.
  2. What was the reasoning for which only men were chosen in this research? is there any explanation?
  3. The authors say: ''The second experimental group 100 (n = 14; age = 20.2 ± 1.2 years; weight = 75.6 ± 25.57 kg; height = 168.4 ± 7.30 cm) underwent classical volleyball classes''.....

    To what refer these courses (classical volleyball program) actually? it is very important for readers to understand these aspects very clearly. The authors must detailed this.

  4. Lines 114-116 - To what questionnaire does it refer? this has not been detailed so far and it is not understood what was the content of this instrument.
  5. The research design must be completed with the necessary information for a very clear understanding of what has been done, these details are not very clearly understood.
  6. Results - it was best in this chapter for the authors to present a comparative analysis for both groups, instead only the results obtained by one group were seen. I think it is imperative that the name of the two groups, i.e control group and experimental group, appear in the title of the table. From the title of the table it is not understood that it refers to both groups, even if this is clearly seen in the content.
  7. As I said before, the questionnaire is not developed, the authors present certain data in the results, but in the methodology it is not understood what was followed, it is difficult to understand or discover what was meant by this tool. The results must be arranged in a logical order, as well the research presentation methodology.
  8. Discussions - this chapter is very little developed and presents little information about the discussion of the results obtained, it is necessary to complete this chapter.
  9. Let's not forget the limits of the paper, they are not presented in this chapter, the authors present in conclusions a phrase but is not clear.
  10. References - in not in accordance with the policy of journal, please see instructions for authors.